# SparseProp: Efficient Event-Based Simulation and Training of Sparse Recurrent Spiking Neural Networks

**Rainer Engelken**
Zuckerman Mind Brain Behavior Institute, Columbia University, New York, USA
Max Planck Institute for Dynamics and Self-Organization, Göttingen, Germany
Göttingen Campus Institute for Dynamics of Biological Networks, Göttingen, Germany
`re2365@columbia.edu`

## Abstract

Spiking Neural Networks (SNNs) are biologically-inspired models that are capable of processing information in streams of action potentials. However, simulating and training SNNs is computationally expensive due to the need to solve large systems of coupled differential equations. In this paper, we introduce *SparseProp*, a novel event-based algorithm for simulating and training sparse SNNs. Our algorithm reduces the computational cost of both the forward and backward pass operations from O(N) to O(log(N)) per network spike, thereby enabling numerically exact simulations of large spiking networks and their efficient training using backpropagation through time. By leveraging the sparsity of the network, *SparseProp* eliminates the need to iterate through all neurons at each spike, employing efficient state updates instead. We demonstrate the efficacy of *SparseProp* across several classical integrate-and-fire neuron models, including a simulation of a sparse SNN with one million LIF neurons. This results in a speed-up exceeding four orders of magnitude relative to previous event-based implementations. Our work provides an efficient and exact solution for training large-scale spiking neural networks and opens up new possibilities for building more sophisticated brain-inspired models.

## 1 Introduction

The cortex processes information via streams of action potentials - commonly referred to as spikes - that propagate within and between layers of recurrent neural networks. Spiking neural networks (SNNs) provide a more biologically plausible description of neuronal activity, capturing in contrast to rate networks membrane potentials and temporal sequences of spikes. Moreover, SNNs promise solutions to the high energy consumption and $CO_2$ emissions of deep networks [1, 2], as well as the ability to transmit information through precise spike timing [3, 4, 5].

Unlike commonly used iterative ODE solvers that necessitate time discretization, event-based simulations resolve neural network dynamics precisely between spike events. This approach ensures machine-level precision in simulations, thereby alleviating the necessity for verification of result robustness with respect to time step size $\Delta t$ and mitigating numerical issues, for instance in the vicinity of synchronous regimes [6]. However, a major drawback of event-based simulations is the computational cost of iterating through all neurons at every network spike time.

To tackle this challenge, we introduce *SparseProp*, an novel event-based algorithm designed for simulating and training sparse SNNs. Our algorithm reduces the computational cost of both the forward and backward pass from $O(N)$ to $O(\log(N))$ per network spike. This efficiency enables numerically exact simulations of large spiking networks and their efficient training using backpropagation through

time. By exploiting network sparsity and utilizing a change of variable to represent neuron states as times to the next spikes on a binary heap, *SparseProp* avoids iterating through all neurons at every spike and employs efficient state updates.

We demonstrate the utility of *SparseProp* by applying it to three popular spiking neuron models. While the current implementation is for univariate neuron models, we discuss a potential extension to more detailed multivariate models later.

Our contributions include:

- Introducing *SparseProp*, a novel and efficient algorithm for numerically exact event-based simulations of recurrent SNNs (section 2 and appendix A for minimal example code).
- Conducting a numerical and analytical scaling analysis that compares the computational cost of our proposed algorithm to conventional event-based spiking network simulations (section 3, Fig 3 and table 1).
- Providing concrete implementations of the algorithm for recurrent networks of leaky integrate-and-fire neurons and quadratic integrate-and-fire neurons (section 4 and here)
- Extending the algorithm to neuron models that lack an analytical solution for the next spike time using Chebyshev polynomials (section 5 and appendix B).
- Extending the algorithm to heterogeneous spiking networks (section 6).

In summary, our proposed *SparseProp* algorithm offers a promising approach for simulating and training SNNs with both machine precision and practical applicability. It enables the simulation and training of large-scale SNNs with significantly reduced computational costs, paving the way for the advancement of brain-inspired models.

## 2 Algorithm for Efficient Event-Based Simulations in Recurrent Networks

We consider the dynamics of a spiking recurrent neural network of $N$ neurons that is described by a system of coupled differential equations [7, 8, 9, 10, 11, 12, 13, 14]:

$$\tau_{\mathrm{m}} \frac{\mathrm{d}V_i}{\mathrm{d}t} = F(V_i) + I_i^{\mathrm{ext}} + \sum_{j,s} J_{ij}\, h(t - t_j^{(s)}). \tag{1}$$

Here, the rate of change of the membrane potential $V_i$ depends on its internal dynamics $F(V_i)$, an external input $I_i^{\mathrm{ext}}$ and the recurrent input $\sum_{j,s} J_{ij}\, h(t - t_j^{(s)})$. $\tau_{\mathrm{m}}$ is the membrane time constant.

Thus, when presynaptic neuron $j$ spikes at time $t_j^{(s)}$ the $i^{\mathrm{th}}$ postsynaptic neuron is driven by by some temporal kernel $h(\tau)$ and coupling strength $J_{ij}$.

For event-based simulations, the dynamics of Eq 1 are evolved from network spike to network spike, usually based on an analytical solution of the membrane potential $V_i(t)$ as a function of time $t$, instead of using iterative ODE solvers like the Runge–Kutta methods. A key aspect of this method for spiking networks is that the spike times are not confined to the time steps of a numerical solver, allowing them to be obtained with machine precision. To simulate the entire network dynamics this way involves four simple steps:

1. Find the next spiking neuron $j^*$ and its spike time $t_{j^*}$.
2. Evolve the state $V_i$ of every neuron to the next network spike time $t_{j^*}$.
3. Update the neurons $i$ postsynaptic to $j^*$ using the weight $J_{ij}$.
4. Reset the state of the spiking neuron $V_{j^*} = V_{\mathrm{reset}}$.

Such event-based simulation schemes are usually used for neuron models that have an analytical solution of the individual neurons' dynamics $V_i(t)$ [10, 14, 15, 16, 17, 18]. The precision of the solution of event-based simulation is then limited only by the machine precision of the computer. In contrast, solving spiking network dynamics with an iterative ODE solver requires asserting that results are not dependent on the integration step size $\Delta t$ of the integration scheme. The total accumulated error usually scale with $\mathcal{O}\left((\Delta t)^p\right)$, where $p$ is the order of the ODE solver. However, both iterating

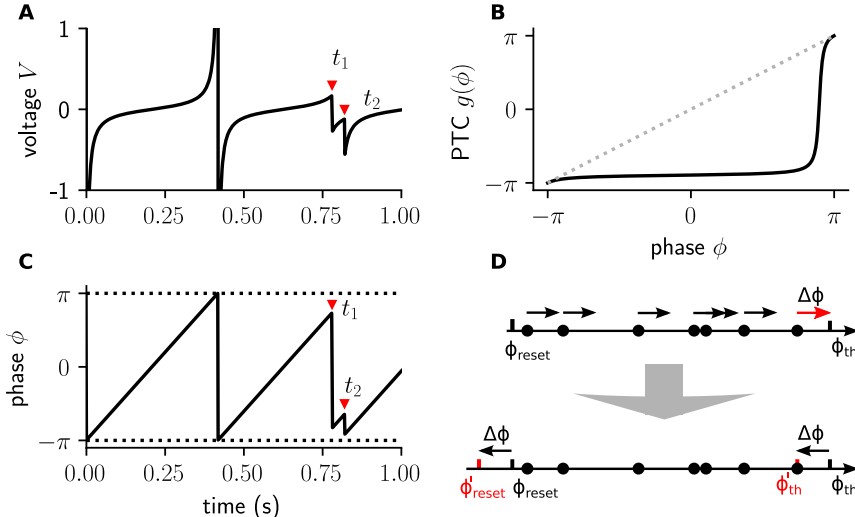

Figure 1: **Change of reference frame allows efficient evolution of network state. A** We map all membrane potentials $V_i(t)$ to phases $\phi_i(t)$ that evolve linearly between spikes with phase velocity $\omega$ [10, 11, 15]. **B** At spike times $t_s$, the phase transition curve (PTC) $g(\phi(t_s))$ has to be evaluated that tells neuron $i$ how its phase $\phi_i$ changes when it receives an input spike. **C** In the phase representation, phases $\phi_i(t)$ evolve linearly in time between spikes, but the amount of change when the neuron receives an input spike depends on its phase. In this example at input spike time $t_1$, the phase change is much larger than at input spike time $t_2$, despite having same size in the voltage representation. **D** In conventional event-based simulations, the phase of all $N$ neurons is shifted at every network spike time by $\Delta\phi_i = \omega\Delta t$, where $\Delta t = (t_{s+1} - t_s)$ is the time to the next network spike. Instead, in *SparseProp* just threshold and reset are changed at every spike.

through all neurons to find the next spiking neuron (step 1) and evolving all neurons to the next spike time (step 2) conventionally require $\mathcal{O}(N)$ calculations per network spike. In the following, we suggest an algorithm that for sparse networks only requires $\mathcal{O}(\log(N))$ calculations per network spike by using a change of reference frame and an efficient data structure. In the next paragraphs, we will describe these two features in more detail.

## 2.1 Change of Reference Frame

For clarity of presentation, we describe our algorithm first for a network of neurons that can be mapped to pulse-coupled phase oscillators like the leaky integrate-and-fire neuron and the quadratic integrate-and-fire neuron (Fig 1 A, C), and discuss later a more general implementation. For pulse-coupled oscillators, the network dynamics can be written as [10, 11, 14, 15, 16, 17, 18, 19]:

$$f\big(\phi_i(t_s)\big) = \begin{cases} g\big(\phi_i(t_s) + \omega(t_{s+1} - t_s)\big) & \text{for } i \in \text{post}(j^*) \\ \phi_i(t_s) + \omega(t_{s+1} - t_s) & \text{else} \end{cases} \tag{2}$$

where $\omega$ is the constant phase velocity and $g(\phi)$ is the phase transition curve (PTC) evaluated for the neurons that are postsynaptic to the spiking neuron $j^*$ just before the next network spike. In this phase representation, in the absence of recurrent input pulses, each phase evolves linearly in time with constant phase velocity $\omega$ from reset phase $\phi^{\text{reset}}$ to threshold phase $\phi^{\text{th}}$ (Fig 1 C). Thus, finding the next spiking neuron in the network $j^*$ amounts for homogeneous networks where all neurons have identical phase velocity to taking the maximum over all phases $\phi_{j^*}(t_s) = \max(\phi_i(t_s))$. The next network spike time $t_{s+1}$ is given by $t_{j^*} = \frac{\phi_{\text{th}} - \phi_{j^*}}{\omega}$. $g(\phi)$ quantifies the phase change of a neuron upon receiving an input spike as a function of its phase (Fig 1 B). In this conventional form of the event-based algorithm [10, 10, 11, 14, 15, 16, 17, 18, 19, 20, 21], all $N$ phases must be linearly evolved to the state immediately before the next network spike time $t_{s+1}$: $\phi_i(t_{s+1}) = \phi_i(t_s) + \omega(t_{s+1} - t_s)$.

This operation, with a computational cost of $\mathcal{O}(N)$, can be avoided in our proposed algorithm. Instead, we introduce a modification where just the threshold and reset phase are changed by a global phase offset $\Delta_\phi$:

$$\Delta_\phi^{(s+1)} = \Delta_\phi^{(s)} + \omega(t_{s+1} - t_s), \tag{3}$$

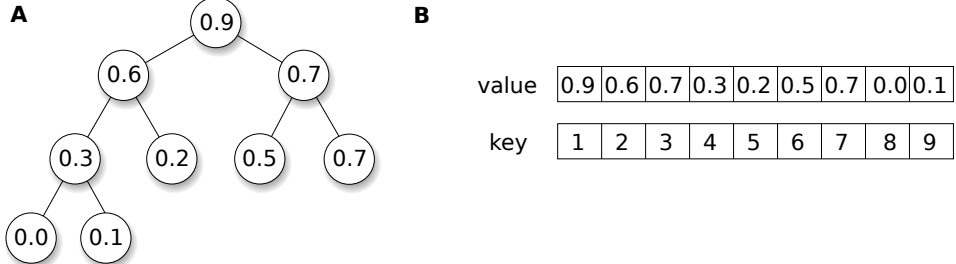

Figure 2: **A suitable data structure allows efficient search for next spiking neuron for numerically exact simulation of large spiking networks. A** By using a priority queue, the next spiking neuron can be found without iteration through all neurons of the network at every spike time. One implementation of a priority queue is a binary heap, as the example shown. In a max-heap, each child node has a value less or equal to its parent node. If an element is deleted, removed, or added, the heap property is restored by swapping parents and children systematically. Finding the node highest priority has a numerical complexity of $\mathcal{O}(1)$ and changing the value of any element has an amortized time complexity of $\mathcal{O}(\log(N))$. **B** Array implementation of binary heap shown in **A**. As per spike all postsynaptic neurons and the spiking neuron have to be updated, this requires on average $K+1$ operation, thus an amortized time complexity $\mathcal{O}(K \cdot \log(N))$ [23].

starting with $\Delta_\phi = 0$ at $t = 0$. Thus $\phi_{s+1}^{\text{th}} = \phi^{\text{th}} - \Delta_\phi^{(s+1)}$. Similarly, $\phi_{s+1}^{\text{reset}} = \phi^{\text{reset}} - \Delta_\phi^{(s+1)}$. See Fig 1D for an illustration. In inhibition-dominated networks, the phases $\phi(t_s)$ will thus become increasingly negative. For long simulations, all phases and the global phase should therefore be reset if the global phase exceeds a certain threshold $\Delta_\phi^{\text{th}}$, to avoid numerical errors resulting from subtractive cancellation due to floating-point arithmetic [22]. We discuss this in more detail in appendix C. In summary, this change of variables reduces the computational cost of propagating the network state from $\mathcal{O}(N)$ to $\mathcal{O}(1)$. Note, that we assumed here identical phase velocity $\omega$ for all neurons, but we will relax this assumption later in section 6.

## 2.2 Efficient Data Structure

We next address the computational bottleneck encountered in very large sparse networks when finding the next spiking neuron, which has a numerical time complexity of $\mathcal{O}(N)$. To overcome this limitation, we introduce an efficient implementation using a priority queue. The priority queue is a data structure that facilitates returning elements with the highest priority and supports insertion and priority updates [24]. We implement the priority queue as a heap-ordered binary tree, where each node's value is greater than its two children's values. This structure enables returning elements with the highest priority in constant time, with a time complexity of $\mathcal{O}(1)$, and inserting elements or updating priorities in logarithmic time, with a time complexity of $\mathcal{O}(\log(N))$ [23]. Thus, only $\mathcal{O}(K \log(N))$ operations have to be performed per network spike, as for $K$ postsynaptic neurons and for one spiking neuron, a new phase needs to be updated in the priority queue.

---

**Algorithm 1** *SparseProp*: Efficient Event-Based Simulation of Sparse Spiking Network

---

1: initialize $\phi(t_0)$, $\Delta_\phi = 0$
2: heapify $\phi(t_0)$
3: warm-up of network $\phi(t_0)$
4: **for** $s = 1 \to t$ **do**
5:     get phase of next spiking neuron: $j, \phi_j = \text{peek}(\phi_i(t_s))$
6:     calculate phase increment: $d\phi = \phi^{\text{th}} - \phi_j(t_s) + \Delta_\phi$
7:     update global phase shift: $\Delta_\phi \mathrel{+}= d\phi$
8:     evaluate phase transition curve: $\phi_{i*}^+(t_s) = g\big(\phi_{i*}^-(t_s) + \Delta_\phi\big)$
9:     reset spiking neuron: $\phi_j(t_{s+1}) = \phi^{\text{re}} - \Delta_\phi$
10:    **if** $\Delta_\phi > \Delta_\phi^{\text{th}}$ **then**
11:       $\phi \mathrel{+}= \Delta_\phi$
12:       $\Delta_\phi = 0$
13:    **end if**
14: **end for**

---

where $\text{peek}(\phi_i(t_s))$ retrieves the highest-priority element without extracting it from the heap. In a min-heap—utilized for the phase representation here —the function fetches the smallest element, representing the next spike time. Conversely, in a max-heap—employed for heterogeneous networks—it obtains the largest element, signifying the maximum phase. The operation generally executes in constant time, $O(1)$, given that the target element resides at the heap's root. Nonetheless, this function leaves the heap structure unaltered.

An example implementation of *SparseProp* in Julia [25] is available here. There are two core innovations in our suggested algorithm: First, we perform a change of variables into a co-moving reference frame, which avoids iterating through all neurons in step 2 (Fig 1). Second, for finding the next spiking neuron in the network, we put the network state in a priority heap (Fig 2). A binary heap has efficient operations for returning the elements with the lowest (highest) key and insertion of new elements or updating of keys [24]. We will next analyze the computational cost of *SparseProp* empirically and theoretically.

## 3 Computational Cost Scaling

Overall, the total amortized cost per network spike of *SparseProp* scales with $\mathcal{O}\left(K \log(N)\right)$. For sparse networks, where $K$ is fixed and only $N$ is growing, the computational complexity of *SparseProp* per network spike thus only scales with $\mathcal{O}\left(\log(N)\right)$.

As the number of network spikes for a given simulation time grows linearly with network size, the overall computational cost for a given simulation time of *SparseProp* scales $\mathcal{O}\left(K\, N\, \log(N)\right)$, which for sparse networks is far more efficient compared to a conventional implementation (see Fig. 3). Specifically, the linearithmic scaling with network size $N$ of our approach, grants remarkable efficiency gains compared to the quadratic scaling of conventional event-based simulations, particularly for sparse networks. For example, simulating a network of $10^6$ neurons and $K = 100$ synapses for 100s takes less then one CPU hour with *SparseProp*, but would take more then 3 CPU years with the conventional algorithm[1]. We provide a detailed comparison of the computational cost in table 1.

## 4 Example Implementation for Integrate-And-Fire Neuron Models

### 4.1 Pseudophase Representation of Leaky Integrate-And-Fire Neurons

For a network of pulse-coupled leaky integrate-and-fire neurons, Eq. 1 reads in dimensionless notation

$$\tau_{\text{m}} \frac{\mathrm{d}V_i}{\mathrm{d}t} = -V_i + I^{\text{ext}} + \tau_{\text{m}} \sum_{j,s} J_{ij}\, \delta(t - t_j^{(s)}) \tag{4}$$

---

[1]These numbers are based on the benchmark on a Xeon® CPU E5-4620 v2 @ 2.60 GHz and 512 GB RAM

|  | conventional algorithm | *SparseProp* |
|---|---|---|
| Find next spiking neuron | $\min_i(\phi_{\text{th}} - \phi_i/\omega)$ | $\text{peek}(\phi_i)$ |
| Evolve neurons | $\phi_i \mathrel{+}= \omega\, dt$ | $\Delta \mathrel{+}= \Delta\phi$ |
| Update postsynaptic neurons | $K$ operations | $K$ operations $+$ $K$ key updates $\mathcal{O}\left(\log(N)\right)$ |
| Reset spiking neuron | one array operation | $\mathcal{O}\left(\log(N)\right)$ |
| Memory cost | $\mathcal{O}\left(N\right)$ | $\mathcal{O}\left(N\right)$ |
| **total amortized costs per network spike** | $\mathcal{O}\left(N+K\right)$ | $\mathcal{O}\left(K\log(N)\right)$ |
| **total amortized costs for fixed simulation time** | $\mathcal{O}\left(N^2\right)$ | $\mathcal{O}\left(K\,N\log(N)\right)$ |

Table 1: **Comparison of computational cost for event-based simulation and training of recurrent spiking networks for different algorithms** $N$ denotes number of neurons, $K$ is the average number of synapses per neuron. For large sparse networks ($K \ll N$) and a fixed number of synapses per neuron $K$, the dominant term grows quadratic for the conventional algorithm [10, 11, 11, 26, 27, 28] and linearithmic ($\mathcal{O}\left(\log(N)\right)$) for the *SparseProp* algorithm.

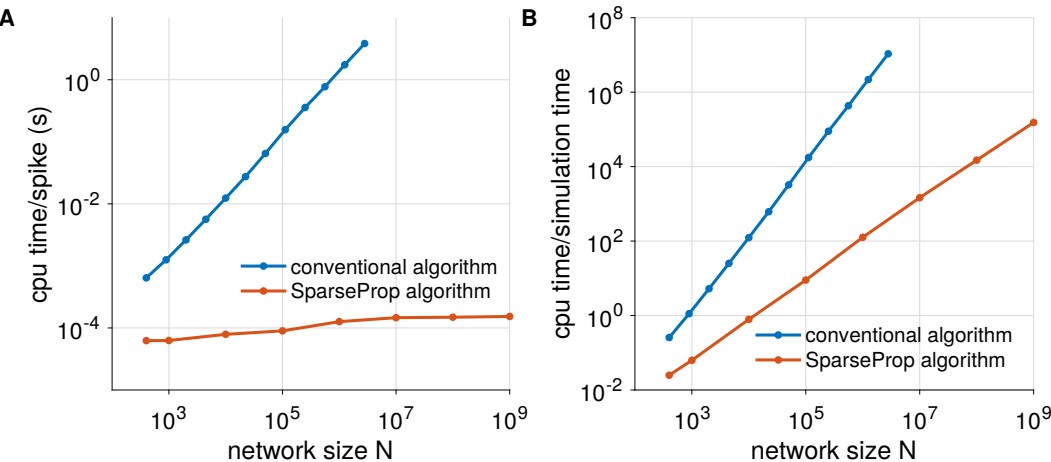

Figure 3: **Benchmark of SparseProp vs. conventional algorithm.** The computational cost per network spike scales linear with network size $N$ in a conventional event-based implementation. In the large sparse network limit, the computational bottleneck is to find the next spiking neuron in the network and to propagate all neurons phases to the next network spike. Both operations can be implemented efficiently. First, instead of shifting all neurons phases, the threshold and reset values can be shifted. Second, by using a binary heap as a data structure for the phases, finding the next phase and keeping the heap-ordering has computational complexity of $\mathcal{O}\left(K\log(N)\right)$. **A**: CPU time per network spike as a function of network size $N$ for an inhibitory network of leaky integrate-and-fire neurons with fixed number of synapses per neuron $K$ [9]. **B**: CPU time per simulation time shows a quadratic scaling for conventional algorithm, but only linearithmic scaling for the novel *SparseProp* agorithm, Benchmark was performed on an Intel® Xeon® CPU E5-4620 v2 @ 2.60 GHz and 512 GB RAM. Parameters: mean firing rate $\bar{\nu} = 1\,\text{Hz}$, $J_0 = 1$, $\tau_{\text{m}} = 10\,\text{ms}$, $K = 100$.

If a membrane potential $V_i$ reaches threshold $V_{\text{th}}$, it is reset to $V_{\text{re}}$. Without loss of generality, we set $V_{\text{th}} = 0$ and $V_{\text{re}} = -1$. Between two network spikes, the solution is given by:

$$V_i(t_{s+1}) = I^{\text{ext}} - \left(I^{\text{ext}} - V_i(t_s)\right)\exp\left(-\frac{t_{s+1} - t_s}{\tau_{\text{m}}}\right) \tag{5}$$

In this pseudophase representation (slightly different from [11, 28]), the phases $\phi_i \in (-\infty, 0]$ describe the neuron states relative to the unperturbed interspike interval.

To obtain the unperturbed interspike interval $T^{\text{free}}$, we have to solve Eq. 5 between reset and threshold in the absence of synaptic input.

$$T^{\text{free}} = -\tau_{\text{m}} \ln \left( \frac{V_{\text{th}} - I^{\text{ext}}}{V_{\text{re}} - I^{\text{ext}}} \right) \tag{6}$$

$$= \tau_{\text{m}} \ln \left( 1 + \frac{1}{I^{\text{ext}}} \right). \tag{7}$$

Its inverse is the phase velocity $\omega = 1/T^{\text{free}}$. The phase $\phi_i$ is thus given by $\phi_i = -\omega \ln \left( \frac{I^{\text{ext}}}{I^{\text{ext}} - V_i} \right)$. The reverse transformation is $V_i = I^{\text{ext}} \left( 1 - \exp \left[ -\frac{\phi_i}{\omega \tau_{\text{m}}} \right] \right)$. Therefore, the phase transition curve is

$$g\big(\phi_{i*}(t_{s+1}^-)\big) = -\omega \ln \big( \exp \big( -\phi_{i*}(t_{s+1}^-)/\omega \big) + c \big), \tag{8}$$

where $c$ is the effective coupling strength $c = \frac{J}{I^{\text{ext}}}$ and $J$ is the synaptic coupling strength taken here to be identical for connected neurons. Usually, as discussed later in the appendix E in the section on balanced networks, $J$ is scaled with $J = \frac{J_0}{\sqrt{K}}$, where $K$ is the number of synapses per neuron. An example implementation of a LIF network with *SparseProp* in Julia [25] is available here.

## 4.2 Phase Representation of Quadratic Integrate-And-Fire Neurons

The quadratic integrate-and-fire (QIF) neuron has, in contrast to the LIF neuron, a dynamic spike generation mechanism and still has an analytical solution $V_i(t)$ between network spikes.

For a network of QIF neurons, Eq. 1 reads in dimensionless voltage representation:

$$\tau_{\text{m}} \frac{\mathrm{d}V_i}{\mathrm{d}t} = V_i^2 + I^{\text{ext}} + \tau_{\text{m}} \sum_{j,s} J_{ij} \, \delta(t - t_j^{(s)}) \tag{9}$$

The quadratic integrate-and-fire model can be mapped via a change of variables $V = \tan(\theta/2)$ to the theta model with a phase variable $\theta \in (-\pi, \pi]$ [29, 30, 31]. The dynamical equation between incoming spikes is the topological normal form for the saddle-node on a limit cycle bifurcation (SNIC) and allows a closed-form solution of the next network spike thanks to the exact derivation of the phase response curve [32]. Therefore, the quadratic integrate-and-fire neuron is commonly used to analyze networks of spiking neurons [10, 33, 34, 35, 36]. When $I_i^{\text{ext}} > 0 \; \forall \; i$, the right-hand side of the dynamics is strictly positive and all neurons would spike periodically in the absence of incoming postsynaptic potentials. In this case, we can choose another particularly tractable phase representation, called phi-representation with $V_i(t) = \sqrt{I^{\text{ext}}} \tan(\phi_i(t)/2)$, where the neuron has a constant phase velocity [10]. This transformation directly yields the phase transition curve

$$g(\phi_i) = 2 \arctan \left( \tan \frac{\phi_i}{2} + c \right), \tag{10}$$

where $c$ is the effective coupling strength $c = \frac{J}{\sqrt{I^{\text{ext}}}}$. The phase velocity is given by $\omega = 2\sqrt{I^{\text{ext}}}$. We considered homogeneous networks here, where all neurons have identical external input and identical membrane time constant, and will later consider the more general case of heterogeneous networks in section 6. An example implementation of a QIF network with *SparseProp* in Julia [25] is available here.

## 5 Event-Based Simulation via Chebyshev Polynomials

*SparseProp* can also be used for univariate neuron models where no analytical solution between spike times is known, e.g., the exponential integrate-and-fire model (EIF)[37, 38]. In this case, the phase transition curve can be calculated numerically before the network simulation. During run time, the phase transition curve and its derivative can be interpolated from the precomputed lookup tables or using Chebyshev polynomials (see appendix B). An example implementation of an EIF network with *SparseProp* in Julia [25] is available here. To test the numerical accuracy of the solution based on a lookup table or Chebyshev polynomials, we compared the spike times of two identical networks of LIF neurons with identical initial condition that were simulated either with the analytical

phase transition curve or using the numerically interpolated solution. We used LIF networks for this comparison, because pulse-coupled inhibitory LIF networks are non-chaotic [11, 21, 39, 40, 41]. Thus, numerical errors originating at the level of machine precision are not exponentially amplified unlike in the chaotic pulse-coupled QIF network dynamics [10]. We show in Fig. 4 that the difference in spike time between the two networks remain close to machine precision and the temporal order of the network spikes is not altered, when replacing the analytical phase transition curve by the

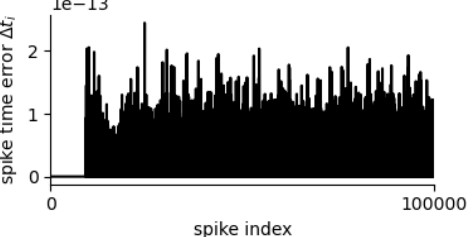 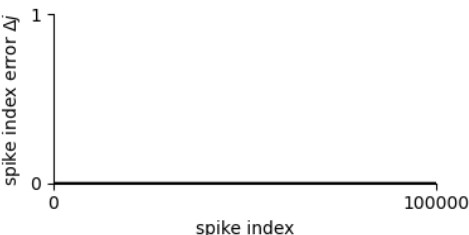

Figure 4: Left: Error of individual spike times for SparseProp are close to machine precision. Right: Error in spike index. Note that despite the small errors in the spike times that are close to machine precision, for non-chaotic network dynamics the spike index is still correct.

## 6 Time-based SparseProp for Heterogeneous Networks

For heterogeneous networks, where there is heterogeneity of external input current $I_i^{\text{ext}}$, or membrane time constant $\tau_i$, or threshold voltages $V_{\text{th}}^i$, every neuron has a different phase velocity $\omega_i$ and a single global phase shift as described in section 2.1 does not suffice.

In that case, we suggest yet another representation. Instead of having a priority queue of phases, instead, we suggest a priority queue of the next unperturbed spike time for all neurons, denoted by $\mathbf{n}(t_0)$, as this can take heterogeneity into account. In this case, the algorithm for network simulation still consists of four steps: First finding the next spiking neuron, then updating the time followed by the update of the postsynaptic neurons by the incoming spikes, finally, resetting the spiking neuron to the reset value. This algorithm also has a numerical complexity of $\mathcal{O}\left(K \log(N)\right)$ per network spike. We provide pseudocode for *SparseProp* in the time-based representation below:

---

**Algorithm 2** *SparseProp*: Event-Based Simulation of Heterogeneous Sparse Spiking Network

---

1: Initialize heterogeneous phase velocity $\omega_i$ based on the neuron model
2: Initialize unperturbed interspike interval $T_i$
3: Initialize time to next spike for all neurons $\mathbf{n}(t_0)$
4: Initialize global time shift $\Delta_t = 0$
5: Heapify $\mathbf{n}(t_0)$
6: Perform warm-up of network state $\mathbf{n}(t_0)$
7: **for** $s = 1 \rightarrow t$ **do**
8:      Get index and spike time of next spiking neuron: $j, n_j = \text{peek}(\mathbf{n}(t_s))$
9:      Calculate time increment: $\mathrm{d}t = n_j(t_s) - \Delta_t$
10:      Update global time shift: $\Delta_t \mathrel{+}= \mathrm{d}t$
11:      Update postsynaptic neurons $i^*$: $n_{i*}^+(t_s) = update((n_{i*}^-(t_s), \Delta_t)$
12:      Reset spiking neuron: $n_j(t_{s+1}) = \Delta_t + T_i$
13:      **if** $\Delta_t > \Delta_t^{\text{th}}$ **then**
14:          $\mathbf{n} \mathrel{-}= \Delta_t$
15:          $\Delta_t = 0$
16:      **end if**
17: **end for**

---

Again, $\text{peek}(\mathbf{n}(t_s))$ retrieves the highest-priority element without extracting it from the heap, which in this case is the minimum over all neurons' next unperturbed spike time. An example implementation of a heterogeneous QIF network with *SparseProp* in this time-based representation is available here.

Note that this can also easily be extended to mixed networks of excitatory and inhibitory neurons (or to $k$ population networks with different external input [42]).

The update of the postsynaptic neurons in line 11 is analogous to the evaluation of the phase transition curve in the homogeneous *SparseProp*, but takes into account the different phase velocities. For example, in the case of a QIF neuron, it takes the form

$$update\big(n_{i*}^-(t_s), \Delta_t\big) = \frac{\pi - g\big(\pi - (n_i - \Delta_t)\omega_i\big)}{\omega_i} + \Delta_t \qquad (11)$$

Here, the phase transition curve, denoted as $g(\phi_i)$, is the same as previously described in Eq. 10: $g(\phi_i) = 2\arctan\left(\tan\frac{\phi_i}{2} + c\right)$ with effective coupling strength $c$. For the QIF neuron, the period $T_i$ is given by $T_i = 2\pi/\omega_i$, with $\omega = 2\sqrt{I^{\text{ext}}}$.

## 7 Efficient Event-Based Training of Spiking Neural Networks

*SparseProp* can also be used for efficient event-based training of spiking neural networks. While surrogate gradients seem to facilitate training spiking networks [43, 44, 45, 46], it was recently claimed that EventProp, which uses exact analytical gradients based on spike times, can also yield competitive performance [47]. To obtain precise gradients for event-based simulations, this algorithm makes use of the adjoint method from optimization theory [48, 49]. In this approach, in the forward pass the membrane potential dynamics (Eq. 1) is solved in an event-based simulation and spike times are stored. In the backward pass, an adjoint system of variables is integrated backward in time, and errors in spike times are back-propagated through the size of the jumps in these adjoint variables at the stored spike times. For event-based simulations of neuron models where the next spike time is analytically known or can be obtained like in section 5, the entries adjoint integration is being taken care of by differential programming [47]. Besides the previously mentioned advantages of event-based simulation regarding precision, this approach is not only sparse in time but also sparse with respect to the synapses, as only synaptic events are used.

We suggest here to further improve the efficiency of event-based training using *SparseProp*. As the backward pass has the identical amortized time complexity of $\mathcal{O}\left(\log(N)\right)$ per network spike as the forward pass, we expect a significant boost of training speed for large sparse networks [47, 50, 51]. Regrettably, the implementation of EventProp presented in [47] was not publicly available during the time of submission. In contrast, the authors of [50] made their re-implementation of EventProp publicly available, but they did not use an event-based implementation but forward Euler to integrate the network dynamics. It will be an important next step to also benchmark the performance of *SparseProp* training. As the gradients are identical, we expect similar results to [47, 50, 51]

## 8 Limitations

In this work, we focus exclusively on recurrent networks of pulse-coupled univariate neuron models such as the quadratic integrate-and-fire neuron, leaky integrate-and-fire neuron, and the exponential integrate-and-fire neuron thereby excluding the exploration of multivariate neuron models. Extending our approach to encompass multivariate models, such as the leaky- or quadratic integrate-and-fire neurons with slow synapses [19, 28, 51, 52], remains a promising direction for future research.

A more fundamental limitation of our study is the apparent incompatibility of event-based simulations with surrogate gradient techniques [46, 53]. While one could introduce 'ghost spikes' in the event-based simulation to emulate surrogate gradients when neurons hit a lower threshold. However, it remains unclear how to preserve the favorable computational scaling of *SparseProp* in this case.

Furthermore, *SparseProp* requires an analytical solution of $V_i(t)$ given the initial state and the neuronal dynamics $\dot{V}_i(t)$ is necessary. This limitation excludes conductance-based models [54, 55, 56], for example. Additionally, time-varying input can only be introduced in the form of deterministic currents or point processes, such as Poisson input. Incorporating stochastic input without an analytical solution of $V_i(t)$ would require further development.

Moreover, the reduction in computational cost per network spike from $\mathcal{O}\left(N\right)$ to $\mathcal{O}\left(\log(N)\right)$ can only be achieved in sparse networks, where the number of synapses per neuron $K$ is much smaller than the

total number of neurons $N$. For dense networks where the number of synapses scales proportionally to the number of neurons [13], a priority queue implemented by a binary heap is disadvantageous compared to a conventional array in the large network limit, as every network spike involves changing the priority of $\mathcal{O}(N)$ neurons thus $\mathcal{O}((N+1)\log(N))$ flops per network spike which corresponds to $\mathcal{O}((N+1)\cdot N\log(N))$ flops for a fixed simulation time. A batch-update of all postsynaptic neurons might be faster for very large networks [57] but this is beyond the scope of this work. It would involve $\mathcal{O}(K+\log(K)\log(N))$ flops corresponding to $\mathcal{O}(N+\log(N)\log(N))$ in dense networks. In the case of purely excitatory sparse spiking networks, a Fibonacci heap might be a more efficient implementation in the large network limit [58], as the 'decrease key' operation takes constant time $\mathcal{O}(1)$ compared to $\mathcal{O}(\log(N))$ in the case of a binary heap. Note that for practical purposes, the asymptotic scaling of the computational complexity of Fibonacci heaps has an unfavorably large prefactor [59]. Therefore, other heap structure implementations might be faster [60]. While large mammalian brains are sparse $N \gg K$, for models of small brains or local circuits with dense connectivity the conventional event-based simulations might be faster.

Additionally, *SparseProp* cannot currently model biophysically detailed neuron features, including dendrites, threshold adaptation, slow synaptic timescales, and short-term plasticity. Incorporating these features would be another valuable avenue for future computational neuroscience research.

Lastly, we did not yet perform a comprehensive systematic benchmark of training with challenging tasks. Such benchmarks would offer further insights into the training performance of our proposed framework.

# 9   Discussion

We introduce *SparseProp*, an efficient and numerically exact algorithm for simulating and training large sparse spiking neural networks in event-based simulations. By exploiting network sparsity, we achieve a significant reduction in computational cost per network spike from $\mathcal{O}(N)$ to $\mathcal{O}(\log(N))$. This speedup is achieved by optimizing two critical steps in the event-based simulation: finding the next spiking neuron and evolving the network state to the next network spike time. First, we employ a binary heap data structure for finding the next spiking neuron and updating the membrane potential of postsynaptic neurons. Second, we utilize a change of variables into a co-moving reference frame, avoiding the need to iterate through all neurons at every spike.

Our results demonstrate the scalability and utility of *SparseProp* in numerical experiments. We demonstrate that *SparseProp* speeds up the simulation and training of a sparse SNN with one million neurons by over four orders of magnitude compared to previous implementations. In contrast to the conventional event-based algorithm, which would require three CPU years to simulate 100 seconds of network dynamics, *SparseProp* achieves the same simulation in just one CPU hour.

This advancement enables the simulation and training of more biologically plausible, large-scale neural network models relevant to theoretical neuroscience. Furthermore, it might pave the way for the exploration of large-scale event-based spiking neural networks in other machine learning domains, such as natural language processing [61], embedded automotive applications [62], robotics [63], and *in silico* pretraining of neuromorphic hardware [1, 64].

The impact of *SparseProp* extends to the recent surge of training for spiking neural networks in theoretical neuroscience [50, 65], machine learning [61, 66, 67], and neuromorphic computing [68, 69, 70]. This algorithm is expected to substantially accelerate the training speed in recent works focused on event-based spiking networks [47, 50, 51].

Future extensions of *SparseProp* to multilayer and feedforward networks would be a promising avenue. Integration of *SparseProp* with recent attempts in biologically more plausible backpropagation and real-time recurrent learning [53, 65, 71, 72, 73, 74] could be used in future research in theoretical neuroscience. To improve training performance, future studies should consider incorporating additional slow time scales, such as slowly decaying synaptic currents [72], and slow threshold adaptation [65], which appear to enhance temporal-credit assignment in recurrent neural networks for rudimentary tasks [53, 65, 72]. Additionally, incorporating gradient shaping and insights from advances in event-based spiking network training [50, 51] could yield further gains in performance.

## Acknowledgments and Disclosure of Funding

I thank S. Goedeke, J. Liedtke, R.-M. Memmesheimer, M. Monteforte, A. Palmigiano, M. Puelma Touzel, A. Schmidt, M. Schottdorf, and F. Wolf for fruitful discussions. Moreover, I thank L.F. Abbott, S. Migirditch, A. Palmigiano, Y. Park, and D. Sussillo for comments on the manuscript. Finally, I thank the Julia community.

Research supported by NSF NeuroNex Award (DBI-1707398), the Gatsby Charitable Foundation (GAT3708), the Simons Collaboration for the Global Brain (542939SPI), and the Swartz Foundation (2021-6). This work was further supported by the Deutsche Forschungsgemeinschaft (DFG, German Research Foundation) 436260547 in relation to NeuroNex (National Science Foundation 2015276) and under Germany's Excellence Strategy - EXC 2067/1- 390729940, DFG - Project-ID 317475864 - SFB 1286, by Evangelisches Studienwerk Villigst, by the Leibniz Association (project K265/2019), and by the Niedersächsisches Vorab of the VolkswagenStiftung through the Göttingen Campus Institute for Dynamics of Biological Networks.

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

# Appendix

## A  SparseProp Code for Efficient Spiking Network Simulation in Julia

This minimal code implements the SparseProp algorithm for simulating a spiking network of $N = 10^5$ LIF neurons.

```julia
using DataStructures, RandomNumbers. Xorshifts, StatsBase, PyPlot

function lifnet(n,nstep,k,j0,ratewnt,τ,seedic,seednet)
        iext = τ*sqrt(k)*j0*ratewnt/1000                  # iext given by balance equation
        ω,c = 1/log(1. + 1/iext),j0/sqrt(k)/(1. + iext)   # phase velocity LIF
        ϕth, ϕshift = 1., 0.                              # threshold for LIF
        r = Xoroshiro128Plus(seedic)                      # init. random number generator
        # initialize binary heap:
        ϕ = MutableBinaryHeap{Float64, DataStructures.FasterReverse}(rand(r,n))
        spikeidx = Int64[]                                #initialize time
        spiketimes = Float64[]                            # spike raster
        postidx = rand(Int,k)

        for s = 1 : nstep                                 # main loop

                ϕmax, j = top_with_handle(ϕ)             # get phase of next spiking neuron
                dϕ = ϕth - ϕmax - ϕshift                 # calculate next spike time
                ϕshift += dϕ                             # global shift to evolve network state
                Random.seed!(r,j+seednet)                # spiking neuron index is seed of rng
                sample!(r,1:n-1,postidx;replace=false)   # get receiving neuron index

                @inbounds for i = 1:k                    # avoid autapses
                        postidx[i] >= j && ( postidx[i]+=1 )
                end
                ptc!(ϕ,postidx,ϕshift,ω,c)               # evaluate phase transition curve
                update!(ϕ,j,-ϕshift)                     # reset spiking neuron
                push!(spiketimes,ϕshift)                 # store spike times
                push!(spikeidx,j)                        # store spiking neuron index
        end
        nstep/ϕshift/n/τ*ω,spikeidx,spiketimes*τ/ω       # output: rate, spike times & indices
end

function ptc!(ϕ, postid, ϕshift, ω, c)                   # phase transition curve of LIF
        for i = postid
                ϕ[i] = - ω*log(exp( - (ϕ[i] + ϕshift)/ω) + c) - ϕshift #(Eq. 12)
        end
end

# set parameters:
#n: # of neurons, k: synapses/neuron, j0: syn. strength, τ: membr. time const.
n,nstep,k,j0,ratewnt,τ,seedic,seednet = 10^5,10^5,100,1,1.,.01,1,1
# quick run to compile code
@time lifnet(100, 1, 10, j0, ratewnt, τ, seedic, seednet);

# run & benchmark network with specified parameters
GC.gc();@time rate,sidx,stimes = lifnet(n,nstep,k,j0,ratewnt,τ,seedic,seednet)

# plot spike raster
plot(stimes,sidx,",k",ms=0.1)
ylabel("Neuron Index",fontsize=20)
xlabel("Time (s)",fontsize=20);tight_layout()
```

# B   Event-Based SparseProp With Numerical Phase Transition Curve

For neuron models where the next spike time cannot be determined analytically and/or the network state cannot be temporally evolved based on a closed-form expression $V(t_{s+t})$, such as the exponential integrate-and-fire model, we present an event-based implementation that has the same computational cost of $\mathcal{O}\left(K \log(N)\right)$ per network spike. To achieve this, we evaluate the phase transition curve either by employing a lookup table or by approximating it using Chebyshev polynomials. We initially describe the implementation based on a lookup table and provide an example implementation:

We begin by numerically integrating the ordinary differential equation that describes the membrane potential $V$ of the neuron model, without considering any external input spikes:

$$\tau_{\mathrm{m}}\frac{\mathrm{d}V}{\mathrm{d}t} = F(V) + I^{\mathrm{ext}}. \tag{12}$$

The integration is performed from the reset voltage $V_{\mathrm{re}}$ to the threshold voltage $V_{\mathrm{th}}$ as follows:

$$V(t) = \int_0^t (F(V) + I^{\mathrm{ext}})/\tau_{\mathrm{m}}. \tag{13}$$

We obtain the numerical solution $V(t)$ by employing the `DifferentialEquation.jl` package with a callback that terminates the integration precisely at $V_{\mathrm{th}}$.

The solution provides us with two crucial pieces of information. Firstly, for each time point $t$, we can now associate a corresponding voltage $V(t)$. Secondly, we also acquire the unperturbed interspike interval $T$, which determines the phase velocity $\omega$. Using the numerically obtained solution $V(t)$, we create a lookup table that maps time to voltage by employing the `DataInterpolations.jl` package. By taking the inverse, we obtain a mapping from a voltage to the next spike time $t(V)$, which we also store as a separate lookup table. The two lookup tables combined give us a numerical phase transition curve $g(\phi)$. We confirmed for the leaky integrate-and-fire neuron and the quadratic integrate-and-fire neuron that this numerical procedure indeed yields the correct phase response curve $d$. Subsequently, we incorporate the numerical phase response curve $d$ into the event-based implementation *SparseProp* implementation and also confirmed the spike times remained identical when comparing with LIF networks with the analytical phase transition curve and identical network realization and identical initial conditions. We used an inhibitory LIF network for this sanity check, because inhibitory pulse-coupled networks of leaky integrate-and-fire neuron are stable with respect to infinitesimal perturbations [11, 21, 39, 40, 41]. This feature in combination with instability with respect to finite-size perturbations renders inhibitory LIF networks as an ideal test-based for numerical precision, because event-based simulations should result exactly in the same spike sequence. This is not the case for networks of QIF neurons, which are typically chaotic [10] (see also [75]).

---

**Algorithm 3** *SparseProp*: Event-Based Simulation Based on Lookup Tables

---
 1:  Set up the ordinary differential equation Eq. 12
 2:  Numerically integrate the ODE from $V_{\mathrm{re}}$ to $V_{\mathrm{th}}$ to obtain $V(t)$.
 3:  Generate a lookup table based on the obtained solution $V(t)$.
 4:  Generate an inverse lookup table based on the solution $t(V)$.
 5:  Combine the lookup table and inverse lookup table to obtain the phase transition curve $g(\phi)$.
 6:  Heapify $\phi(t_0)$
 7:  Perform the warm-up of network $\phi(t_0)$
 8:  **for** $s = 1 \to t$ **do**
 9:      Get index and phase of next spiking neuron: $j, \phi_j = \mathrm{peek}(\phi_i(t_s))$
10:      Calculate phase increment: $d\phi = \phi^{\mathrm{th}} + \Delta - \phi_j(t_s)$
11:      Update global phase shift: $\Delta \mathrel{+}= d\phi$
12:      Evaluate phase transition curve: $\phi_{i^*}^+(t_s) = g\left(\phi_{i^*}^-(t_s) + \Delta\right)$
13:      Reset spiking neuron: $\phi_j(t_{s+1}) = \phi^{\mathrm{re}} - \Delta$
14:      **if** $\Delta > \Delta^{\mathrm{th}}$ **then**
15:          $\phi \mathrel{+}= \Delta$
16:          $\Delta = 0$
17:      **end if**
18: **end for**

---

An example implementation of an EIF network using *SparseProp* in Julia, in the balanced state, is available here.

As an alternative to generating two lookup tables to numerically define the phase transition curve $g(\phi)$ using `DataInterpolations.jl`, we propose the use of Chebyshev Polynomials for a direct estimation of $d$. We provide an example implementation using `ApproxFun.jl` here.

## C   Numerical precision of SparseProp

Since *SparseProp* employs event-based simulation with analytical expressions for calculating the next spike times, the algorithm operates with a precision limited only by machine epsilon. Numerically, over many spike events, the global phase offset $\Delta\phi$ incrementally increases. For sufficiently large values of $\Delta\phi$, catastrophic cancellation can arise. For instance, when $\Delta\phi = 10^6$ and this is subtracted from a very small $d\phi$. Under these conditions, the issue is that the adjacent floating-point number greater than $10^6$ in double-precision arithmetic differs by approximately $2^{-30} \approx 1.16 \times 10^{-10}$. One can mitigate this numerical error by resetting both the global phase shift and the phases of individual neurons upon exceeding a specified threshold. This counteracts the subtractive cancellation errors inherent to floating-point calculations. Nonetheless, such resets are seldom required and occur at intervals on the order of $cN$ steps. The coefficient $c$ is a substantial prefactor contingent on the tolerable floating-point error; for an error of $2^{-30}$, $c$ would be $10^6$. This can be verified in Julia with the command `log2(nextfloat(1.0e6)-1.0e6)`.

## D   Online Sparse Erdős–Rényi Connectivity Matrix Generation

In order to benchmark the simulation of large networks using the *SparseProp* algorithm in Fig 3, we implemented online generation of the sparse Erdős–Rényi connectivity matrix. This approach allows us to generate the set of postsynaptic neurons deterministically online, without the need to store the entire weight matrix. Storing the entire adjacency matrix for networks with $10^9$ neurons would be impractical due to its prohibitively large size, even when using the *compressed sparse column* format. By using this online generation method, we only require memory to store the state of all $N$ neurons. For a network of $10^9$ neurons, this corresponds to approximately 64bit/double float $\times 10^9$neurons $\times$ 8bytes/bit $= 7.4$ GB of RAM. Our implementation draws inspiration from previous works [76, 77]. It is important to note that the example code provided in appendix A has a fixed indegree, similar to the approach in [9]. Here, we present an example of online sparse adjacency matrix generation with truly Erdős–Rényi connectivity, where the indegree follows a binomial distribution.

Code example:

This code for Julia v1.9 implements a sparse Erdős–Rényi connectivity matrix for efficiently simulating a spiking network.

```julia
using Random, DataStructures, RandomNumbers. Xorshifts, StatsBase

# before simulation:
N, K, j, seedNet = 10^6, 10^2, 2, 1

rng = Xoroshiro128Star(seedNet) # init. random number generator (Xorshift)
npostAll = Array{UInt16,1}(undef, N)
for n = 1:N
        npostAll[n] = UInt16(randbinom(rng, N, K / N))
end

# before simulation when neuron j spikes:
npost = npostAll[j]
# spiking neuron index is seed of rng
Random.seed!(rng, j)
# get postsynaptic neuron index
postIdxes = sample(rng, 1:N-1, npost, replace = false)
# avoid autapses
for np = 1:npost
        if postIdxes[np] >= j
                postIdxes[np] += 1
        end
end

# before simulation, the following function was used
function randbinom(rng, n::Integer, p::Real)
        log_q = log(1.0 - p)
        x = 0
        sum = 0.0
        while true
                sum += log(rand(rng)) / (n - x)
                sum < log_q && break
                x += 1
        end
        return x
end
```

# E    Spiking Networks in the Balanced State

In order to provide a comprehensive understanding and ensure the accessibility of our work, we now introduce the concept of the balanced state, which motivates the scaling of the synaptic coupling strength with $1/\sqrt{K}$. Neural activity in cortical tissue has typically asynchronous and irregular pattern of action potentials [78, 79], despite the fact that individual neurons can respond reliably [80, 81, 82, 83]. This is commonly explained by a balance of excitatory and inhibitory synaptic currents [84, 85], which cancels large mean synaptic inputs on average. A dynamically self-organized balance can be achieved without the fine-tuning of synaptic coupling strength in heterogeneous networks, if the connectivity is inhibition-dominated and the couplings are strong such that a small active fraction of incoming synapses can trigger an action potential [86]. The statistical properties of this state are described by a mean-field theory, which is largely insensitive to the specific neuron model [87].

In our study, we investigated large sparse networks consisting of $N$ neurons arranged on a directed Erdős–Rényi random graph with a mean degree of $K$. All neurons $i = 1, \ldots, N$ received constant positive external currents $I^{\text{ext}}$ and non-delayed $\delta$-pulses from the presynaptic neurons $j \in \text{pre}\,(i)$.

The values of the external currents were chosen using a bisection method to achieve a target average network firing rate $\bar{\nu}$.

## F  Setup of the Balanced Networks

The coupling strengths in our networks were scaled with $1/\sqrt{K}$, such that the magnitudes of the input current fluctuations had the same order of magnitude in all studied networks. Assuming that inputs from different presynaptic neurons exhibit weak correlations, which is justified in balanced network [13, 88], the collective input spike train received by neuron $i$ can be effectively modeled as a Poisson process with a rate denoted as $\Omega_i = \sum_{j \in \mathrm{pre}(i)} \nu_j \approx K\bar{\nu} \equiv \Omega$. Here, $\bar{\nu}$ represents the average firing rate of the network, and $K$ denotes the average number of presynaptic neurons. Assuming the compound input spike train follows a Poisson process, the autocorrelation function of the input current can be expressed as follows:

$$
\begin{align}
C(\tau) &= \langle \delta I(t) \delta I(t+\tau) \rangle_t \tag{14}\\
&\approx \left(\frac{J_0}{\sqrt{K}}\right)^2 \Omega \int \delta(t-s)\delta(t+\tau-s)\mathrm{d}s \tag{15}\\
&= \frac{J_0^2}{K}\Omega\delta(\tau) \tag{16}\\
&\approx J_0^2\bar{\nu}\delta(\tau) \tag{17}
\end{align}
$$

where $\delta$ denotes the Dirac delta function. In the diffusion approximation, characterized by high Poisson rate and weak coupling, fluctuations in the input currents can be described as Gaussian white noise with a magnitude given by

$$
\sigma^2 = J_0^2\bar{\nu}. \tag{18}
$$

Note that due to the scaling of the coupling strengths $J = -\frac{J_0}{\sqrt{K}}$ with the square root of the number of synapses $K$ the magnitude of the fluctuations $\sigma^2$ is independent of the number of synapses. Therefore, the input fluctuations do neither vanish nor diverge in the thermodynamic limit and the balanced state in sparse networks emerges robustly [86, 87].

The existence of a fixed point of the population firing rate in the balanced state for large $K$ follows from the equation of the network-averaged mean current:

$$
\bar{I} \approx \sqrt{K}(I_0 - J_0\bar{\nu}). \tag{19}
$$

In the limit of large $K$, self-consistency requires the balance between excitation and inhibition, specifically $I_0 = J_0\bar{\nu}$. If $\lim_{K\to\infty}(I_0 - J_0\bar{\nu}) > 0$, the mean current $\bar{I}$ would diverge to $\infty$, resulting in neurons firing at their maximum rate. The resulting strong inhibition would break the inequality, leading to a contradiction. On the other hand, if $\lim_{K\to\infty}(I_0 - J_0\bar{\nu}) < 0$, the mean current $\bar{I}$ would diverge to $-\infty$, causing the neurons to be silent. Again, the lack of inhibition breaks the inequality. The self-consistency condition in the large $K$ limit can only be satisfied when

$$
I_0 - J_0\bar{\nu} = \mathcal{O}\left(\frac{1}{\sqrt{K}}\right),
$$

which ensures that the excitatory external drive and the mean recurrent inhibitory current cancel each other. Note that since $I_0 - J_0\bar{\nu} = \mathcal{O}(1/\sqrt{K})$ the network mean current has a finite value in the large $K$-limit. The average population firing rate, expressed in units of the membrane time constant $\tau_{\mathrm{m}}^{-1}$, can be approximated as

$$
\bar{\nu} = \frac{I_0}{J_0} + \mathcal{O}\left(\frac{1}{\sqrt{K}}\right). \tag{20}
$$

This approximation generally becomes exact for large $K$. For excitatory-inhibitory mixed network, an analogous self-consistency argument can be made which results in a set of inequalities that must be fulfilled to achieve a balanced state [87].

## G  Code Availability

An example implementation of networks of LIF, QIF and EIF neurons using *SparseProp* in Julia, in the balanced state, is available at https://github.com/RainerEngelken/SparseProp.

