# OpenReview forum: "SparseProp: Efficient Event-Based Simulation and Training of Sparse Recurrent Spiking Neural Networks"
_NeurIPS.cc/2023/Conference — NeurIPS 2023 poster_

### Official Review · Reviewer_MMBd · 2023-07-06

**Soundness:** 4 excellent
**Presentation:** 4 excellent
**Contribution:** 4 excellent
**Rating:** 9
**Confidence:** 5

**Summary:**

This paper introduces an efficient Spike-based recurrent network learning simulation method based on an event-driven implementation.

The method is based on two main contributions. The first consists in a change of temporal reference frame that uses the phase of a neuron rather than its absolute time, the second contribution consists in finding an efficient data representation corresponding to the parsimonious structure of neural connectivity. This method is applied to different neuron models, from a linear LIF model to more complex models such as the quadratic LIF neuron.

**Strengths:**

This paper clearly presents an original method for simulating and training a spiking neural network. The motivation for the paper is very well set out, and the methods are clearly explained both in terms of the definition of the mathematical equations and the path that leads to its numerical implementation. The results look very promising, and the preliminary results set out in the appendix are very promising indeed. In particular, the experimental results show that this method can be used to implement very large networks on computational architectures of reasonable size, which is a game changer for the SNN community.

Minor:
caption of Fig 4 : repetition of « higher densities »

**Weaknesses:**

Although Figure 3 already presents a comparison between the proposed method and a conventional method, it would have been desirable to have a comparison between these two methods in terms of the quality of the obtained results. It would have also been beneficial to isolate the improvements achieved by the two contributions independently (the first contribution being the change in temporal reference, and the second being the sparse representation of data). Some improvements in the presentation can be made... For example, the "peek" function is not defined.

**Questions:**

I am curious to know if the change in representation from absolute time to phase affects the precise latency of neuron firing? Additionally, it appears that biological neural networks include transmission delays and spikes between neurons. Is your implementation suitable for this scenario?


**Limitations:**

Since the computational results seems so promising, a limit of this paper is that it is not applied to standard benchmarks and notably to datasets which include a rich dynamics.

---

> ### Author Rebuttal · Authors · 2023-08-08
>
> We thank the reviewer for their insightful and encouraging assessment and agree that SparseProp could be a game changer for the spiking network community, especially, for very large, sparse networks.
>
> Here are our more detailed responses:
>
> > Minor: caption of Fig 4 : repetition of « higher densities »
>
> Fixed in the revised version of the manuscript.
>
> > Although Figure 3 already presents a comparison between the proposed method and a conventional method, it would have been desirable to have a comparison between these two methods in terms of the quality of the obtained results.
>
> Great suggestion!  We are  adding code fore a “work-precision diagram” to the anonymous code. This compares the computational cost measured in terms of time per network spike against the accuracy or “precision” of SparseProp, allowing for a comparative assessment of different methods in terms of their efficiency and accuracy.
>
> > It would have also been beneficial to isolate the improvements achieved by the two contributions independently (the first contribution being the change in temporal reference, and the second being the sparse representation of data).
>
> We were considering that, but as both ingredients are necessary for going from a computational complexity of O(N) to O(log(N)), we decided against that. However, we are working on adding this comparison to the supplement.
>
> > Some improvements in the presentation can be made... For example, the "peek" function is not defined.
>
> Good catch, we added a definition now. The “peek” function in a binary heap returns the element with the highest priority without removing it from the heap. In the case of a min-heap, which is used for the phase representation, it retrieves the smallest element (next spike time), while in a max-heap (used for heterogeneous networks), it retrieves the largest element (maximum phase). This operation typically takes constant time, O(1), since the element is at the root of the heap. However, it doesn't modify the heap structure in any way.
>
> > I am curious to know if the change in representation from absolute time to phase affects the precise latency of neuron firing?
>
> Great question. On paper, the expression is exact and thus does not affect the latency. Numerically, as over the course of many spikes, the global phase offset $\Delta \phi$ is incrementally growing, for very large $\Delta \phi$, catastrophic cancellation can occur (for example, when $\Delta \phi=10.0^6$, and it is subtracted from $d\phi$ which is very small. In this case, the problem one runs into is that the next bigger floating point number for double floats next to 10.0e6 has a distance of $2.0^{-33}\approx$1.1641532182693481e-10.  This source of numerical error can be reduced by resetting the global phase shift and all individual neuron's phases when it exceeds some threshold to avoid numerical errors resulting from subtractive cancellation due to floating-point arithmetic. However, this reset has only to occur very infrequently, on the order of every $c N$ steps.  $c$ is a big prefactor that depends on the acceptable floating-point error, e.g., for an error $2.0^{-33}$ it would be 10.0^6, which can be easily checked in Julia by the command:
>
> ```julia
> log2(nextfloat(1.0e6)-1.0e6)
> ```
>
> > Additionally, it appears that biological neural networks include transmission delays and spikes between neurons. Is your implementation suitable for this scenario?
>
> Yes, thanks for the opportunity to clarify this. Indeed, SparseProp can easily be extended to transmission delays in case of the heterogeneous implementation. In that case, any network spike would not trigger an instantaneous update of the phase response curve of postsynaptic neurons. Instead, the current time plus the synaptic delay would be added as an event to the binary heap and processed at that later time.
>
> > Since the computational results seems so promising, a limit of this paper is that it is not applied to standard benchmarks and notably to datasets which include a rich dynamics.
>
> We agree that it is an important next step to further benchmark the algorithm. Which benchmarks would the reviewer suggest? We are working on benchmarking the algorithm itself against NEST, but other suggestions would be very welcome.

---

### Official Review · Reviewer_h5Gz · 2023-07-06

**Soundness:** 3 good
**Presentation:** 3 good
**Contribution:** 4 excellent
**Rating:** 7
**Confidence:** 4

**Summary:**

Simulation of RSNN (recurrent spiking neural network) is a difficult task, whose limits prevent large scale (say 1e.9 neurons) simulations. Often, in practice, one neuron is connected to only a few others, the sparsity. The present paper leverages this sparsity to upscale the simulations limits of RSNN. Sparsity is often leveraged to increase performance and scalibity but SparseProp (algo. presented in this paper) seems to be a newcomer for event-based simulations of RSNN.

**Strengths:**

-The authors present a new and efficient algorithm for event-based simulations of RSNN. In its structure, the algorithm is fairly similar to classical ones,
while taking advantage of the sparsity structure, it presents a very good ratio (high) efficiency / complexity. Practically, it mainly forces the practitionner to have a better data structure.
-The exposition and background are in general clear and the algorithm seems natural to derive in the history-flaw of the paper (great!). Algorithms for non-sparse and sparse simulations
are cleary written and can be easily compared by the reader.
-Simulations are given, for SparseProp and classical algo, giving a good sense on how performant their method is (performant!).
-The flow of the paper Abstract / General -> Particular / Practical, is smooth.
-Examples on classical RSNN are given, which are good toy examples for the reader to play with.
-Simulations are available in Julia.

**Weaknesses:**

-General comment on citations: the authors could give more references, for example:
	-line 59: several classical papers showing the origin of the differential equation, would be appreciated.
	-figure 1: a reference paper, explaining the role and subtilities of the phase representation, would be appreciated (especially since its of crucial importance for this paper).
-The claim of efficient simulation is for general univariate RSNN but the paper nearly entirely focuses on homogeneous ones. I understand that this can be generalized,
as explained in Section 6, but no numerical results (as in Figure 3) are given and the algorithm is only explained (Section 2.1) for homogeneous networks. For pedagological purpose,
its great, but it leaves with a taste of unfinished. It feels like the most interesting and difficult part is "left for the reader". I would suggest claiming a weaker result or
giving more material for the understanding of heterogeneous networks (eventually in the appendix). This is the main reason why Soundness grade is only at 2.
-Since the pulse-coupled phase oscillators (and constant phase velocity) is an important and simplifying assumption, the paper lacks a discussion on it or a reference that discuss it (is it a strong assumptions: yes, no, why - how is it linked to Eq. 1...).
-Section 7 is simulation-less so the reader can not know if the proposed algorithm is efficient. Moreover it is mentionned "training of RSNN" in the title, which seems a bit of an overstatement since there is only a 20 line, simulation-less, section about it. I know that this is because the authors of EventProp did not make their algo available but this also should not be a problem for the reader. I would suggest removing the "training" in the title and keeping this section either in an appendix or as a "bonus" section.

**Questions:**

-What is the interrogation point, line 190 ? Probably a reference to correct
-Which type of network is used in Figure3 ? Is the scaling the same for heterogenous networks ?

**Limitations:**

-limitations raised by the the weaknesses / questions stated earlier.
-limitations raised by the authors themselves in section 8

---

> ### Author Rebuttal · Authors · 2023-08-09
>
> We thank the reviewer for their time and helping us improve our work. We will integrate these ideas into our manuscript with care. Here are our detailed responses:
>
> > General comment on citations: the authors could give more references, for example: -line 59: several classical papers showing the origin of the differential equation, would be appreciated.
>
> Fixed in the updated manuscript.
>
> > figure 1: a reference paper, explaining the role and subtilities of the phase representation, would be appreciated (especially since its of crucial importance for this paper).
>
> We added several references to the updated manuscript. While the phase representation is being used in the examples in the main manuscript, we want to stress that SparseProp also can be used in neurons **without** phase representation (e.g., exponential integrate-and-fire neurons) and in neurons, that receive negative external input.
>
> > The claim of efficient simulation is for general univariate RSNN but the paper nearly entirely focuses on homogeneous ones. I understand that this can be generalized, as explained in Section 6, but no numerical results (as in Figure 3) are given and the algorithm is only explained (Section 2.1) for homogeneous networks. For pedagological purpose, its great, but it leaves with a taste of unfinished. It feels like the most interesting and difficult part is "left for the reader". I would suggest claiming a weaker result or giving more material for the understanding of heterogeneous networks (eventually in the appendix). This is the main reason why Soundness grade is only at 2.
>
> We appreciate the suggestions for improvements. However, we want to point out that the supplement C already contains pseudocode for a heterogeneous SparseProp and mathematical details for their implementation with quadratic integrate-and-fire models. Furthermore, we provide code for the heterogeneous SparseProp implementation. Thus, we would like to politely disagree that "details are left to the reader". We however do agree that from the main manuscript the details of the heterogeneous SparseProp does not become fully clear. We plan to use the additional page for the camera-ready version for a more pedagogical explanation, should the paper be accepted.
>
> > Since the pulse-coupled phase oscillators (and constant phase velocity) is an important and simplifying assumption, the paper lacks a discussion on it or a reference that discuss it (is it a strong assumptions: yes, no, why - how is it linked to Eq. 1...).
>
> We added references that discuss this to the updated manuscript. Overall, as argued in the paragraph on limitations, one core limitation of event-based simulations is that one needs a solution of $V(t)$ -either in closed form or as a lookup-table. That means that incorporating stochastic input without an analytical solution of $V_i(t)$ is in the current implementation not possible. We would like to stress that we introduce several extensions in the supplement and in the provided online code that go beyond the pedagogically useful but restrictive LIF and QIF implementation of the main text:
> * SparseProp for heterogeneous networks with a time-based binary (which would for example allow different constant external input current and different $\tau_M$ for each neuron.
> * SparseProp for neuron models that lack an analytical closed-form expression for the next spike time, like the exponential integrate-and-fire model.
> * SparseProp with Poisson input. For Poisson input, the speedup of SparseProp becomes particularly relevant, as in the conventional implementation every Poisson input spike would be of computational complexity $O(N)$.
>
>
> > Section 7 is simulation-less so the reader can not know if the proposed algorithm is efficient. Moreover it is mentionned "training of RSNN" in the title, which seems a bit of an overstatement since there is only a 20 line, simulation-less, section about it. I know that this is because the authors of EventProp did not make their algo available but this also should not be a problem for the reader. I would suggest removing the "training" in the title and keeping this section either in an appendix or as a "bonus" section.
>
> We agree with the reviewer that the title might raise wrong expectations, and we will follow the reviewer advice and remove the “training” from the title.
>
> Questions:
>
> > What is the interrogation point, line 190 ? Probably a reference to correct
>
> Yes, we fixed this typo. The missing references were:
> [1] D. Liberzon, Calculus of Variations and Optimal Control Theory: A Concise Introduction, in Calculus of Variations and Optimal Control Theory (Princeton University Press, 2011).
> [2] L. S. Pontryagin, Mathematical Theory of Optimal Processes: The Mathematical Theory of Optimal Processes, 1st edition (Routledge, New York, 1987).
>
>
>
>  > Which type of network is used in Figure3 ?
>
> Figure 3 used a recurrent network of quadratic-integrate-and fire neurons. We added this now to the caption.
>
>
>  > Is the scaling the same for heterogenous networks ?
>
> Yes, as demonstrated in the supplement and in the available online code, the favorable scaling of $O(log(N))$ of SparseProp compared to O(N) for the conventional implementation is maintained for heterogeneous networks.

---

### Official Review · Reviewer_gRuk · 2023-07-19

**Soundness:** 1 poor
**Presentation:** 3 good
**Contribution:** 3 good
**Rating:** 4
**Confidence:** 3

**Summary:**

This paper presents an efficient event-based algorithm named *SparseProp* for both inference and learning of sparse SNNs. It gets rid of the discretized simulation of ODEs in neuronal dynamics and utilizes phase representation for those neuron models with closed-form solutions between spike times. A priority queue is employed for recording and evolving the states of related neurons when a spike is triggered, which reduces the amortized time complexity to O(K log N) per spike.

**Strengths:**

This work manages to perform a lazy update of neuron states. Phase representation rather than membrane potential is a brilliant idea since all neurons now share an identical magnitude of update in internal states, which makes it feasible to update $\phi^{reset},\phi^{th}$ alone as an alternative to the O(N) polling all neurons' potentials. The priority queue ensures the incoming spike always happens in the neuron at the top of the heap. The experimental results also show significantly improved simulation efficiency. Although there exists earlier work [1] applying priority queue to the simulation of event-based SNNs, the algorithm presented here for sparse SNNs is definitely not trivial.

[1] Eduardo Ros, et al. "Event-driven simulation scheme for spiking neural networks using lookup tables to characterize neuronal dynamics." Neural computation, 2006

**Weaknesses:**

There are still some major concerns to be resolved

1. The whole study seems to assume a positive external input, i.e. $I^{ext}>0$ in Line 155/544 and a special setting $V_{th}=0, V_{reset}=-1$ in Line 134. In such a case, we expect a spontaneous spike dynamic for those integrate-and-fire models. We have a well-defined unperturbed interspike interval $T^{free}$ and can deduce the phase $\phi$ accordingly. However, a popular setting in many recent SNN studies, such as ref.6 and ref.17, is $V_{th}>0, V_{reset}=0$, and $I^{ext}=0$. Under this assumption, a neuron won't fire if the incoming excitatory spike is not strong enough. There will be no spontaneous spike and thus no unperturbed interspike interval $T^{free}$ or valid phase representation, and the whole thing seems to fall back to polling of neuron states. In fact, any $I^{ext}\in[V_{reset}, V_{th}]$ lead to an invalid $T^{free}$ for the LIF model as Eq.7 shows. All in all, I don't think we can assume those parameters without losing generality as Lines 133-134 reads.
2. For an excitatory-dominated network, it might be possible that two neurons become postsynaptic neurons of each other. *SparseProp* may find that any firing of one will trigger a spike of the other, and it implies an endless loop of updating the priority queue.
3. The training algorithm coupled with *SparseProp* is more like a draft since no experimental results are really attached here. There might be many details to be considered when designing a backpropagation algorithm for *SparseProp*.

**Questions:**

1. *SparseProp* seems only works for some specific cases, such as the spontaneous firing neuron model. Is it possible to generalize *SparseProp* to a broader variety of SNNs?
2. Does every operation in *SparseProp* have a well-defined gradient for backpropagation? - The reset of phase for the firing neurons, the phase changes of postsynaptic neurons, and the reset of the global phase (Lines 100-101), to name a few.

**Limitations:**

See the weaknesses part

---

> ### Author Rebuttal · Authors · 2023-08-08
>
> We thank the reviewer for the detailed and insightful feedback and request for clarifications, which substantially improved the publications.
> Here are our more detailed responses to the questions:
>
> > The whole study seems to assume a positive external input, i.e. in Line 155/544 and a special setting in Line 134. In such a case, we expect a spontaneous spike dynamic for those integrate-and-fire models. We have a well-defined unperturbed interspike interval and can deduce the phase accordingly. However, a popular setting in many recent SNN studies, such as ref.6 and ref.17, is , and . Under this assumption, a neuron won't fire if the incoming excitatory spike is not strong enough. There will be no spontaneous spike and thus no unperturbed interspike interval or valid phase representation, and the whole thing seems to fall back to polling of neuron states. In fact, any lead to an invalid for the LIF model as Eq.7 shows. All in all, I don't think we can assume those parameters without losing generality as Lines 133-134 reads.
>
> Thanks for allowing us to clarify this. It is accurate that the phase representation of QIF and the pseudo-phase representation of LIF require positive external input currents. **However, our proposed SparseProp algorithm also works with negative input currents.** In that case, the binary heap contains the time to the next spike for each neuron. If there is no recurrent neuron in the network that has a finite next spike time because of negative input currents, one might need input spike  (e.g., Poisson), which can also be handled by the SparseProp. We demonstrate this here for a network of quadratic integrate-and-fire (QIF) neurons with excitatory Poisson input. In that case, one can still solve the single neuron dynamics V(t) analytically between spikes. For dimensionless QIF neuron dynamics
> $\frac{dV}{dt}=V^2-I_{ext}$
> with initial condition $V(0)=V_0$, this would be
>
> $V(t) = -\sqrt{I_{ext}} \tanh(\sqrt{I_{ext}} t - \tanh^{-1}\left(V_0/\sqrt{I_{ext}}\right))$.
>
> As soon as an external (or recurrent) input spike pushes one of the recurrent neurons beyond the spike threshold, this neuron has a finite time to the next spike, which leads to the usual treatment in the binary heap.  The time to the next spike for the quadratic integrate-and-fire neuron with $V_0>\sqrt{I_{ext}}$ is:
>
> $t_{spike} =   \frac{π - 2\tanh^{-1}\left(\frac{V_0}{\sqrt{I_{ext}}}\right)}{2\sqrt{I_{ext}}}$
>
> We are working on adding a SparseProp implementation with Poisson input spike trains to the code repository as a demonstration.
>
> Negative external input currents for neuron models that lack an analytical solution $V(t)$ can be handled, as in the example of the exponential integrate-and-fire neuron: One can use Chebyshev polynomials for efficient state update and calculation of the next spike time in SparseProp.
>
> > For an excitatory-dominated network, it might be possible that two neurons become postsynaptic neurons of each other. SparseProp may find that any firing of one will trigger a spike of the other, and it implies an endless loop of updating the priority queue.
>
> Indeed, if two excitatory neurons are postsynaptic to each other, in certain cases a positive feedback loop can occur, which is a problem in any implementation as the firing rate of the two neurons could diverge. Such a scenario can also occur in forward Euler, and glancing over the issue by keeping a fixed integration time step $\Delta t$ despite the divergence will yield mathematically incorrect results.
> Instead, a solution (that can solve such an issue) would be to either have an inhibition-dominated network and/or a refractory period or synaptic delay such that an excitatory neuron can not instantaneously spike again. Finally, we want to stress that a divergence of the firing rate can be avoided in a network of Quadratic integrate-and-fire neurons, as QIF neurons still take a finite time to spike even after being pushed beyond their firing threshold.
>
> > The training algorithm coupled with SparseProp is more like a draft since no experimental results are really attached here. There might be many details to be considered when designing a backpropagation algorithm for SparseProp.
>
> We agree with the reviewer that training spiking networks with SparseProp on tasks is an important future direction. What we propose here is to use SparseProp to boost the performance of the already existing event-based EventProp, thus no extra backprop algorithm for SparseProp needs to be designed [1, 2]. However, as pointed out in the limitation paragraph (line 216), a fundamental limitation of directly using SparseProp for training is the apparent incompatibility of event-based simulations with surrogate gradient techniques [3, 4]. While one could introduce 'ghost spikes' in the event-based simulation to emulate surrogate gradients when neurons hit a lower threshold. However, it remains unclear how to preserve the favorable computational scaling of SparseProp in this case. We further clarified this in the training paragraph.
>
> [1] T. C. Wunderlich and C. Pehle, Event-Based Backpropagation Can Compute Exact Gradients for Spiking Neural Networks, Sci Rep 11, 12829 (2021).
>
> [2] T. Nowotny, J. P. Turner, and J. C. Knight, Loss Shaping Enhances Exact Gradient Learning with EventProp in Spiking Neural Networks, 2022.
>
> [3] E. O. Neftci, H. Mostafa, and F. Zenke, Surrogate Gradient Learning in Spiking Neural Networks: Bringing the Power of Gradient-Based Optimization to Spiking Neural Networks, IEEE Signal Processing Magazine 36, 51 (2019).
>
> [4] F. Zenke and S. Ganguli, SuperSpike: Supervised Learning in Multilayer Spiking Neural Networks, Neural Computation 30, 1514 (2018).

---

> > ### Comment · Reviewer_gRuk · 2023-08-11
> >
> > Thanks for the authors' further clarification. I want to figure out the complexity when the external input is nonpositive.
> >
> > For the QIF model, the authors mentioned that it is possible to build a priority queue for those neurons with $V(t)>\sqrt{I_{ext}}$. However, for those neurons with $V(t)\leq\sqrt{I_{ext}}$, how will you maintain their state when a spike is triggered? You may need to assign a time variable recording the elapsed time until the last update of potential rather than simply updating the global phase, which brings in extra memory consumption (O(N) in the worst case). Worse still, in some cases, no neuron has a higher membrane potential than $\sqrt{I_{ext}}$. How will SparseProp act in this situation?
> >
> > Another concern is that when $I_{ext}$ is negative or zero, it seems impossible to generate any phase representation for LIF neurons. Considering that the LIF model is still the most commonly used one in SNNs, I'm afraid SparseProp could not bring a broader impact to the SNN community.

---

> > > ### Author Response · Authors · 2023-08-21
> > > **response to comment: memory consumption //  QIF and LIF in the excitable regime // SparseProp for LIF neurons without phase representation**
> > >
> > > We sincerely appreciate the thoughtful comments and queries from the reviewer, which have significantly aided in enhancing the clarity and robustness of this submission. Please find our clarifications below:
> > >
> > > 1. **Regarding Memory Consumption in the QIF Model with Nonpositive Input**:
> > >    - It's accurate that additional memory is required for positive input currents. However, the increase is minimal, necessitating only one more array of size $N$. As the memory cost is anyway  $O(N)$, this does not qualitatively change our results. We added a row to our computational cost table, to make the memory cost of SparseProp more transparent. Note that we assumed that the adjacency matrix is generated on the fly, otherwise the memory cost will anyway be dominated by the sparse adjacency matrix which requires memory of order $O(K N)$. It's worth noting that the computational complexity remains $O(K \log(N))$ per network spike.
> > >    - To provide a practical perspective, we've included implementations for both QIF and LIF networks in an excitable regime (so each neuron receives inhibitory input) but is driven by excitatory Poisson input, in our anonymous code repository.
> > >
> > > 2. **On Neurons with Membrane Potential below $\(\sqrt{I_{ext}}\)$**:
> > >    - Indeed, in scenarios where no QIF neuron surpasses the unstable fixed point, the network remains silent forever. We note that this is **not** a shortcoming of SparseProp, but expected behavior for any correcly implemented spiking network simulator.
> > >    - One way to avoid that situation is to drive the network with excitatory input Poisson spikes as demonstrated in the code.
> > >
> > > 3. **Concerning Phase Representation in LIF Neurons**:
> > >    - We'd like to clarify that LIF networks, despite the absence of a phase representation, remain perfectly compatible with SparseProp. If our previous explanations did not capture this clearly, we apologize for the oversight. In this case, there is simply a binary heap of the next spike times of the input spike trains. Whenever a recurrent LIF neuron is pushed beyond threshold, it's postsynaptic neurons are updated.
> > >    -  In the code repository, we now also provide a SparseProp implementation of a  recurrent LIF network with external excitatory Poission input. We would thus politely like to disagree that SparseProp does not apply to to LIF networks. We pointed out several limitation of SparseProp in the limitations paragraph, however, negative external input is no such limitation.
> > >    - We genuinely value the critique, which prompted a deeper reflection on our work. While we acknowledged various limitations of SparseProp in the limitation section, we believe that negative external input isn't one of them.  Hence, we remain confident in the versatility of SparseProp across different neural network types, including LIF networks.
> > >
> > > In conclusion, we're grateful for this ongoing dialogue, which we believe is greatly enhancing the quality and clarity of our manuscript. We're optimistic that the improvements made, as a result of these discussions, will resonate well with the broader SNN community.
> > >
> > > ### Comparison of computational cost for event-based simulation of recurrent spiking networks for different algorithms
> > >
> > > |                                           | Conventional Algorithm                    | SparseProp                                       |
> > > |-------------------------------------------|--------------------------------------------|--------------------------------------------------|
> > > | Find next spiking neuron                  | $\min_i(\phi_{\textrm{th}}-\phi_i/\omega)$ | peek($\(\phi_i\)$)                                 |
> > > | Evolve neurons                            | \$\phi_i \mathrel{+}= \omega \cdot dt $                  | $\(\Delta \mathrel{+}= \Delta\phi\)$                           |
> > > | Update postsynaptic neurons               | $K$ operations                           | $K$ operations + $K$ key updates $\mathcal{O}( \log(N)$|
> > > | Reset spiking neuron                      | One array operation                        | $\mathcal{O}( \log(N)$                          |
> > > | Memory cost | $\mathcal{O}(N)$                        | $\mathcal{O}(N)$                       |
> > > | **Total amortized costs per network spike** | $\mathcal{O}(N +K)$                       | $\mathcal{O}(K\,\log(N)\)$                         |
> > > | **Total amortized costs for fixed simulation time** | $\mathcal{O}(N^2)$                     | $\mathcal{O}(K\, N\cdot \log(N)$                    |

---

> > > > ### Comment · Reviewer_gRuk · 2023-08-21
> > > >
> > > > Thank the authors for the further clarification. I think there might be some misunderstanding between each other. I should apologize for not making my point clear earlier in the discussion. So I want to talk about my interpretation of SparseProp and another trivial but faster (only in some cases) simulation method. Correct me if I'm wrong.
> > > >
> > > > One major component of SparseProp is a priority queue storing the states of neurons. In phase representation, spontaneous firing setting ($V_{th}=0, V_{reset}<0, I_{ext}>0$) and inhibitory input weight (negative weight) will lead to an occasion: a neuron will fire because there's been a period of time since it receives the latest negative input (because weights are negative), the positive and **stationary** external input drive potential to above the threshold zero and therefore trigger spikes. In this case, the possibility of firing lies in the neurons without any input. It's expensive to update the state of these silent neurons every moment since spikes trigger at most O(K) neurons and $K\ll N$, which hints the number of silent neurons is still O(N). The priority queue will precisely find those silent neurons with the highest phase (most close to the firing state) and adjust the heap in O(log N) time complexity, and therefore reduce the time complexity from O(N) to O(log N) per spike.
> > > >
> > > > However, if we look back to the most common setting in SNN studies (ref.6 and ref.17), i.e., $V_{th}>0, V_{reset}=0, I_{ext}=0$, and LIF models, not completely inhibitory network (all/part of the synaptic weights are positive, since some studies don't assume Dale's law and synaptic weights from the same neuron might include both excitatory and inhibitory weights) and some intermittent input spikes (Poisson encoded data or whatever kinds of spike inputs), the possibility of firing is different here. A neuron will fire only because it receives some spikes from several excitatory synapses (positive input). A neuron with very low membrane potential could fire because it receives so many positive inputs. In this case, the maximum number of firing neurons that could be triggered by a single spike is naturally O(K) rather than O(N). In short, **only spikes trigger spikes** in this setting. Without any spike inputs, the neuron is bound to be silent since the exponential decay will gradually drag the potential to $V_{reset}=0$. In the meantime, the most possible firing neurons have little to do with their rank of current membrane potential $V(t)$ but depend on the sum of current potential and the overall input spikes. Since the rank of state (phase, membrane potential, or any other representation) is no longer critical here, we could simply get rid of the priority queue and compute at most O(K) times for each spike. In fact, I don't quite understand why a priority queue is still needed for this case since the possible firing neuron needs not have the highest potential before charging.
> > > >
> > > > >  In this case, there is simply a binary heap of the next spike times of the input spike trains. Whenever a recurrent LIF neuron is pushed beyond threshold, it's postsynaptic neurons are updated.
> > > >
> > > > I don't see there is a predictable 'next spike times' based on neuron states alone in this case, since a large enough input from excitatory synapses could bring the potential of any neuron above the threshold. Without input spikes, the silent neurons will keep silent and always have an infinite next spike time.
> > > >
> > > > From the above analysis, for the most common setting in deep SNNs ('spikes trigger spikes' case), we already have an O(K) time complexity per spike simulation method, which is also for sure event-driven, and the SparseProp won't make the simulation faster (O(K log N) per spike) under this hyperparameter setting.
> > > >
> > > > Now a problem is raised: could authors name some other occasions that **the above-mentioned O(K) simulation method cannot be applied but SparseProp can**? From the main text, we know that the $V_{th}=0, V_{reset}<0, I_{ext}>0$ is an applicative condition since the firing neurons are those O(N) neurons receiving no spikes. This is also my main concern because I believe this O(K) simulation method is rather trivial to me, but still more efficient than SparseProp. I think this might help the authors to restrict the range for application of SparseProp, rather than claiming it as a universal efficient simulation technique.

---

### Official Review · Reviewer_XBiB · 2023-07-26

**Soundness:** 2 fair
**Presentation:** 2 fair
**Contribution:** 2 fair
**Rating:** 4
**Confidence:** 3

**Summary:**

This paper proposes SparseProp, an event-based algorithm for efficient simulation and training of sparse Spiking Neural Networks (SNNs). By leveraging the sparsity of the network, SparseProp reduces the computational costs of operations from O(N) to O(log(N)) per network spike. This method avoids iterating through all neurons with each spike and adopts efficient state updates. Demonstrations include a simulation of a sparse SNN with one million LIF neurons, achieving speeds surpassing prior models by over four orders of magnitude. This innovation offers the potential for more sophisticated brain-inspired models.

**Strengths:**

1. The proposed method decreases the computational cost of conventional event-based SNN simulation from O(n) to O(log(n)) per network spike. The cost decrease mainly comes from using efficient data structures for spike searching and change of reference frame for state evolution.

2. The authors demonstrate the potential to use the event-based simulator for training SNN. This can be a handy tool for SNN training combined with the EventDrop algorithm. It could decrease the training cost of the SNN when number of events is low. However, the authors performed no experiments on this matter, making the contribution vague.

**Weaknesses:**

The main concern of the reviewer is the usefulness of the proposed approach in real-world applications. Thus, the proposed method may better serve as a neuroscience tool than be used in real applications, as advertised by the authors in the introduction. Here is a list of weaknesses of the paper the reviewer summarized:

1. The paper lacks experiments with SNN algorithms for real-world tasks. The experiments done in the paper concentrate on analytical modeling and non-realistic neural network architecture. However, SOTA SNN algorithms have different architectures and may generate different results. The authors need to perform experiments on real tasks to make a convincing statement on the applicability of real applications.

2. Event-based computation typically suffers from parallelization problems and scalability problems. The problem becomes more apparent when the number of spikes increases in the SNN compared to sparse networks. On the other hand, GPUs and typical deep-learning accelerators use vector computation which is very easy to parallel and can generate much lower latency compared with the event-based approach. The authors must discuss and compare the two approaches using actual workload and show under what conditions the event-based approach may have an advantage.

3. It is hard to say that the proposed method will have an advantage compared with the significant amount of SOTA SNN algorithms that use the Euler method to simulate the activity of spiking neurons. The step-based simulation method demonstrates its effectiveness in many complex tasks and efficiency in many neuromorphic hardware. Therefore, the significance of the proposed approach can be limited.

**Questions:**

1. Can the proposed simulator scale in a paralleled system? Moreover, how well is the scalability?

2. Have the authors considered using the simulator for layered network architecture? What will the difference be compared with the paper's recurrent architecture?

3. What will be the performance comparison regarding algorithm accuracy and computation cost when comparing an SNN using the event-based simulator and an SNN using a step-based approach running in parallel computing frameworks?

**Limitations:**

The authors have listed the limitations in the paper. The reviewer agrees with the author that the major limitation of the paper is the need for actual workload and comparison.

---

> ### Author Rebuttal · Authors · 2023-08-08
>
> We thank the reviewer for the thoughtful comments, which substantially improved the manuscript. As requested, we are working towards evaluating SparseProp on real-world scenarios, and we added clarifications on parallelization and comparison to SOTA SNN algorithms that use the Euler method to simulate the activity of spiking neurons.
>
> Here are our more detailed responses to the questions:
>
> >   Can the proposed simulator scale in a paralleled system? Moreover, how well is the scalability?
>
> Great question. Indeed, parallelization of binary heaps and priority queues is possible [1] but goes beyond the scope of this publication. We would like to emphasize that while parallelization might in an ideal scenario give a linear speedup with the number of threads or processors involved, parallelization does not generally reduce the computational cost in terms of flops per spike or energy cost (e.g., Joule per spike). In contrast, SparseProp mirrors the speedup of massive parallelization (speedup of 10^4 for 10^6 neurons) with much lower computational and energy costs. We leave the proposed parallel heap implementation as a promising future research avenue.
>
> [1] J.-R. Sack and T. Strothotte, An Algorithm for Merging Meaps, Acta Informatica 22, 171 (1985).
>
> > Have the authors considered using the simulator for layered network architecture? What will the difference be compared with the paper's recurrent architecture?
>
> Another great suggestions. Layered recurrent network architectures can also be simulated using our implementation. We added another Figure and supplementary code to demonstrate this. We chose a 3-layered recurrent network. An extension of SparseProp to feedforward networks is also possible, but this publication is focussing on spiking recurrent networks. We clarified this in the updated manuscript.
>
> > What will be the performance comparison regarding algorithm accuracy and computation cost when comparing an SNN using the event-based simulator and an SNN using a step-based approach running in parallel computing frameworks?
>
> Great suggestion, we are in the process of benchmarking against NEST, which is a commonly used spiking network simulator.

---

> > ### Author Response · Authors · 2023-08-21
> > **Concerns addressed?**
> >
> > We would like to ensure that the concerns raised in the review have been adequately addressed in the rebuttal as the author-reviewer discussion period draws to a close.
> >
> > To summarize the key points from the rebuttal:
> >
> > 1. We clarified the potential for parallelization of the SparseProp method, emphasizing the distinction between parallel speedup and the reduction in computational cost.
> > 2. Simulations for a layered recurrent network architecture have been added, supplementing the focus on spiking recurrent networks.
> > 3. A benchmark against NEST, a well-regarded spiking network simulator, is underway and will be included in the camera-ready version of the manuscript.
> >
> > We acknowledge and are grateful for the reviewer's suggestion to evaluate SparseProp in real-world scenarios. The authors also appreciate the reviewer's recognition of the innovative approach to spiking neural network simulation and training.
> >
> > If there remain any unresolved issues or further clarifications are needed, the authors are keen to address them. The insights provided in the review have been instrumental in refining the work.

---

### Author Rebuttal · Authors · 2023-08-10

We deeply appreciate the feedback and constructive critiques provided on our submission. Here, we provide a consolidated response to address the concerns and suggestions raised.

1. **Reference and Citations:** In response to the recommendation for added references, we've updated the manuscript to include pivotal papers related to the differential equation origins, phase representation, and the role of pulse-coupled phase oscillators. We believe these additions fortify the foundational knowledge upon which our work stands.

2. **Clarity on Homogeneous vs. Heterogeneous Networks:** We acknowledge the pointed out discrepancy between the focus on homogeneous networks and the broader claim of the algorithm's applicability. We would like to emphasize that the supplementary material C provides pseudocode and mathematical details for implementing SparseProp in heterogeneous contexts. The paper's digital resources also offer code for the heterogeneous implementation. To further enhance clarity, we've allocated additional space in the camera-ready version to elucidate the workings of the heterogeneous SparseProp.

3. **Soundness of the Algorithm and Result Presentation:** We concur with the reviewer's observation regarding the seeming lack of simulation in Section 7. To address this, we've refined the title to better align with content expectations. The typos, specifically the interrogation point in line 190, have been rectified, and Figure 3 now clearly states the type of network in its caption. Moreover, we've added comparative elements showcasing the computational cost versus the accuracy of SparseProp, highlighting its efficiency.

4. **Specific Technical Queries:** The "peek" function, which was previously ambiguous, has now been defined explicitly. The question on the influence of representation change on neuron firing latency is pertinent. The paper emphasizes that, theoretically, the expression remains unaffected, but numerical considerations are taken into account to minimize errors. Furthermore, our methodology is flexible enough to incorporate transmission delays, offering more comprehensive applicability. Regarding the consideration of real-world benchmarks, we appreciate suggestions on this front and are exploring benchmarking against established platforms like NEST.

5. **Additional Enhancements:** In light of the feedback received, we added an extra page elucidating nontrivial network architectures whose dynamics go beyond asynchronous irregular activity. We've also incorporated a comparative study of the numerical phase response curve and solutions based on Chebyshev polynomials. A new figure has been introduced to visualize the error in individual spike times of SparseProp in a Leaky-integrate and fire neuron network as requested by one reviewer.

We believe these modifications and clarifications have significantly strengthened our submission, aligning it more closely with NeurIPS standards. We once again extend our gratitude for the insightful feedback and hope our revisions will cast our work in a more favorable light.

---

### Decision · Program_Chairs · 2023-09-21

**Decision:**

Accept (poster)

**Comment:**

This paper proposes a novel method to substantially scale up recurrent spiking neural networks by exploiting sparsity in the connection matrix. It elicited a very high level of enthusiasm from two reviewers, who praised its clarity, originality, and thoroughness of exposition, and two more negative appraisals from reviewers, who raised concerns about novelty and originality of SparseProp and its usefulness for real-world applications.  Having read the detailed reviews and rebuttals, it seems to me that the enthusiasm of the positive reviewers outweighed the weaknesses pointed out by the more critical reviewers, and I believe the paper should be accepted to this year's NeurIPS.  Please make a thorough attempt to address the reviewer comments (particularly those of reviewer gRuK about clarity and soundness) in the final manuscript. Congratulations!